# Low Prevalence of Chronic Obstructive Pulmonary Disease in Greenland—A Call for Increased Focus on the Importance of Diagnosis Coding

**DOI:** 10.3390/ijerph20095624

**Published:** 2023-04-24

**Authors:** Andreas Brix, Kristine Flagstad, Marie Balslev Backe, Michael Lynge Pedersen, Maja Hykkelbjerg Nielsen

**Affiliations:** 1Queen Ingrid’s Primary Health Care Center, 3900 Nuuk, Greenland; 2Steno Diabetes Center Greenland, 3900 Nuuk, Greenland; 3Greenland Center for Health Research, Institute of Health and Nature, University of Greenland, 3905 Nuuk, Greenland; 4Steno Diabetes Center Copenhagen, 2730 Herlev, Denmark; 5Department of Clinical Medicine, Aarhus University, 8000 Aarhus, Denmark

**Keywords:** chronic obstructive pulmonary disease, prevalence, quality of care, Inuit, Greenland

## Abstract

The aim of this study was to estimate the prevalence of patients diagnosed with chronic obstructive pulmonary disease (COPD) according to age, gender, and residence in Greenland and to investigate the associated quality of care. The study was performed as an observational cross-sectional study using data on patients diagnosed with COPD, extracted from the electronical medical record (EMR) in Greenland. The total prevalence of patients aged 20–79 years diagnosed with COPD in Greenland in 2022 was 2.2%. The prevalence was significantly higher in the capital Nuuk compared to the remaining parts of Greenland (2.4% vs. 2.0%, respectively). Significantly more women than men were diagnosed with COPD, but the lung function of men was found to be significantly reduced/impaired compared to women. The prevalence of patients aged 40 years or above was 3.8%. The quality of care was significantly higher among patients living in Nuuk compared to the remaining parts of Greenland for eight out of ten quality indicators. The prevalence of COPD in Greenland is lower than in other comparable populations and might be underestimated. Continued focus on early detection of new cases and initiatives to improve and expand monitoring of quality-of-care measurements, including both additional clinical and patient reported outcomes, are recommended.

## 1. Introduction

Chronic obstructive pulmonary disease (COPD) is an immense health issue worldwide with severe impacts on quality of life. Claiming 3.2 million peoples’ lives in 2019, it is the third leading cause of death worldwide [1]. The current 2019 Global Initiative for Chronic Obstructive Lung Disease (GOLD) strategy document defines COPD as a common, preventable, and treatable disease characterized by persistent respiratory symptoms and airflow limitations [2]. Globally, the prevalence of COPD varies greatly [3,4]. In the Western world, approximately 50% of individuals with COPD remain undiagnosed due to insufficient diagnostic setup and utilization of spirometry [5], particularly in general practice [3]. Smoking is the most common risk factor in the Western world. With 55% of the Greenlandic population aged 15–94 years smoking on a daily basis [6], a high prevalence of COPD in Greenland is expected.

In Greenland, former studies have documented low diagnostic activity of COPD in the period from 2010 to 2019 [7,8,9]. A Greenland study from 2021 found a prevalence of patients using medication targeting obstructive lung disease at 7.5%, but only 152 patients were diagnosed with COPD, corresponding to 0.4% of the population [7]. Furthermore, the study showed that among patients treated with medication targeting obstructive lung disease, only 26% were tested with spirometry within two years [7]. Contributing factors to the low use of spirometry may be associated with lack of diagnostic focus and the use of diagnosis coding in the Greenland electronical medical record (EMR), which has been limited to hospitalized patients in Nuuk as well as patients with diabetes. Additionally, problems with spirometry equipment including access to disposable mouthpieces may also be an issue, as well as a probably low awareness in the population. 

Since 2008, the national prevalence of diabetes and associated quality of care has been monitored continuously through a lifestyle initiative, which, based on mandatory diagnosis coding, has resulted in a high quality of care [10]. In 2020, the Steno Diabetes Center Greenland (SDCG) was established to strengthen and further develop the management of lifestyle-related diseases including diabetes, COPD, and hypertension, with focus on diagnosis coding and quality of care. By January 2021, it became mandatory for all medical doctors at Queen Ingrid’s Primary Health Care Center in Nuuk to add a medical diagnosis code to all contacts with the health care in the EMR. In 2021 and 2022, SDCG has summoned all Nuuk-living patients aged 40 years or above receiving medication targeting obstructive lung disease to a spirometry, thereby establishing a baseline of patients diagnosed with COPD using spirometry to improve diagnostics and treatment of COPD in Greenland. 

The aim of this study was to estimate the prevalence of patients diagnosed with COPD and the associated quality of care in Greenland in 2022 and to investigate possible differences in age and gender as well as regional differences. 

## 2. Materials and Methods

### 2.1. Study Design

The study was performed as an observational cross-sectional study using data from the EMR in Greenland in 2022. 

### 2.2. Setting

Greenland is the largest island in the world, with an area of more than 2 million km^2^. Due to the icecap, most of the population lives along the coastline. The capital Nuuk is populated by approximately 19,000 people, roughly one-third of the population of Greenland [11].

The remaining part of the population lives in 16 minor towns and around 50 smaller settlements. The health care system in Greenland is organized into five regions, each with a regional hospital, whereas each town has a primary health care center, and the settlements operate with smaller health care units [12]. The health care system provides free prescription of medicine for all citizens, and furthermore, all services are free of charge.

Nuuk has a unit for primary health care, Queen Ingrid’s Primary Health Care Centre located next to Queen Ingrid’s Hospital. Queen Ingrid’s Hospital is the only hospital in Greenland responsible for secondary specialized health care. Throughout Greenland, patients diagnosed with COPD are treated at a regional hospital or primary health care center according to SDCG guidelines. Since 2020, all consultations at SDCG have encompassed diagnosis coding according to the ICPC-2 code [13,14]. 

### 2.3. Study Population and Variables

The study population includes all residents aged 20 years or above diagnosed with COPD. The background population was the population of Greenland by 1 July 2022, aged 20 years or above, except Tasiilaq, a town located on the east coast of Greenland. The population was extracted from Greenland Statistics’ online statistics bank [11].

Medical data were extracted from the EMR from all parts of Greenland except for Tasiilaq, where the EMR has not yet been fully implemented due to slow internet connections. The inclusion criterion for patients included in the study was the registration of a medical diagnosis code for COPD: the ICPC-2 code; R95 and/or ICD10 codes; DJ440, DJ441, or DJ449 in the EMR by 19 October 2022.

Information extracted from the EMR included age, weight, height, blood pressure (home blood pressure if any, if not office blood pressure), spirometry data, COPD assessment (CAT) score, and smoking status. BMI was calculated based on the patients’ height and weight excluding shoes and outerwear, using the formula BMI = kg/m^2^. Spirometry was performed by the use of EasyOne Air [15]. Spirometry is offered to patients who have symptoms of lung disease or are being treated for undiagnosed lung disease or come to the health care center for check-ups for COPD or asthma or prior to major operations on the lungs. Spirometry data included forced expiratory volume in 1 s (FEV1), forced vital capacity (FVC), and the FEV1/FVC ratio. Furthermore, data on expected FEV1 (%) and expected FVC (%) were included based on European ethnicity as reference. 

To estimate the age-specific prevalence of patients diagnosed with COPD, patients were divided into age groups of 20 years. Quality of care was described according to selected available international indicators for COPD management in primary health care [16,17] and selected key performance indicators set for SDCG (see Table 1) [13].

The indicators for quality of care comprised smoking status, blood pressure, nutritional status, physical activity, COPD grade, debut time of COPD, CAT score, influenza vaccination, pneumococcus vaccination, and spirometry. 

Smoking status was evaluated within the previous year. Information on quantity and former smoking (pack years) was not available in this study. Blood pressure was defined as the latest registered measurement within one year. Nutritional status was defined as the latest registered height and weight within the previous year. The definition of physical activity was the latest registration of physical activity within the previous two years. COPD grade was defined as the latest registration of a COPD grade (A, B, C, or D) within the previous two years. Debut time of COPD was defined as a registration of the debut time of COPD or not. CAT score was defined as a registered CAT score. Influenza vaccine was defined as the latest registered influenza vaccine within the previous two years. The influenza vaccine is an annual vaccine, but we chose it as a two-year indicator since it is more stable over time and not influenced by minor random circumstances such as delayed delivery of flu vaccines in Greenland. The definition of pneumococcus vaccine was a registered vaccine against pneumococcus or not. Spirometry was defined as the latest registered measurement or not measured within two years. For spirometry, the data represent the best post- or prebronchodilator measurement, as it was not possible to retrieve information regarding whether a postbronchodilator had been taken or not.

### 2.4. Statistical Analysis

Estimates were calculated with 95% confidence intervals (CI). Normally distributed parameters were described using mean and standard deviation (SD). The check for normal distribution was performed using QQ-plot. Normally distributed variables were compared using *t*-tests. Proportions were compared using chi-square tests. When calculating the age- and gender-specific prevalence of patients with COPD, patients were divided into age groups of 20 years. Statistical analysis was performed in R version 4.1.2. *p*-values below 0.05 were considered significant.

The study was approved by the Science Ethics Committee in Greenland (No. 2022–15) and by the Agency of Health and Prevention in Greenland. All data were handled anonymously.

## 3. Results

In 2022, a total of 884 patients with a mean age of 64 years were diagnosed with COPD. Of these, 53.2% (470) were females and 46.8% (414) were males. Patients aged 40–79 years constituted more than 99% of the patients diagnosed with COPD. 

### 3.1. Basic Characteristics

Table 2 shows the basic characteristics of the study population according to gender. 

When comparing men and women, men were significantly heavier and had a higher diastolic blood pressure (*p* < 0.001). Furthermore, men had a lower force expiratory volume within 1 s (FEV1 (%)) and force vital capacity (FVC (%)) in percentage of the expected, compared to women, thus resulting in a significantly reduced lung function (FEV1/FVC-ratio (%)) (*p* < 0.001). Among the 884 patients, a total of 539 patients were current smokers (65.3%).

### 3.2. Prevalence

Table 3 presents the prevalence of patients diagnosed with COPD according to age, gender, and residence.

The total prevalence of patients diagnosed with COPD aged 20–79 years was 2.2%, and the prevalence among seniors aged 40 years or above was 3.8%. The total prevalence was significantly higher among patients living in the capital Nuuk (2.4%) compared to patients living in the remaining parts of Greenland (2.0%) (*p* = 0.006). The prevalence of seniors diagnosed with COPD was also significantly higher among patients living in Nuuk (4.5%) compared to patients living in the remaining parts of Greenland (3.4%) (*p* < 0.001). Furthermore, the prevalence of patients diagnosed with COPD was significantly higher in Nuuk compared to the remaining parts of Greenland for patients aged 40–59 years (2.4% vs. 1.6%, *p* < 0.001), 60–79 years (8.9% vs. 5.9%, *p* < 0.001), and 80 years or above (12.6% vs. 5.9%, *p* = 0.009).

When looking at gender, significantly more women than men were diagnosed with COPD (2.4% vs. 1.9%, *p* = 0.001). The prevalence of patients aged 40 years or above was also higher among women compared with men (4.3% vs. 3.3%, *p* < 0.001). Furthermore, in the 40–59 age group, significantly more women than men were diagnosed with COPD (2.4% vs. 1.5%, *p* < 0.001).

### 3.3. Quality of Care

Table 4 shows the specific quality of care indicators for patients diagnosed with COPD in 2020 and 2022. Moreover, the quality-of-care indicators were compared between patients living in Nuuk and the remaining parts of Greenland, as well as between men and women.

The quality of care according to eight out of ten process indicators was significantly higher among patients in Nuuk compared to patients from the remaining parts of Greenland (*p* < 0.05). In accordance, more patients in Nuuk had smoking status assessed (72.5% vs. 57.9%, *p* < 0.001) and blood pressure measured (59.1% vs. 47.9%, *p* = 0.001) within the previous year compared to patients from the remaining parts of Greenland. Furthermore, a significantly higher proportion of patients in Nuuk had their nutritional status measured within the previous year compared to patients from the remaining parts of Greenland (74.7% vs. 68.1%, *p* = 0.032). When looking at process indicators within the previous two years, more patients in Nuuk had a spirometry performed and a COPD grade registered compared to patients from the remaining parts of Nuuk (92.9% vs. 78.3%, *p* < 0.001 and 60.2% vs. 16.5%, *p* < 0.001, respectively). A debut time of COPD was registered in 53.6% of the patients living in Nuuk compared to only 13.9% of the patients living in the remaining parts of Greenland (*p* < 0.001). A CAT score was registered in 40.7% of the patients living in Nuuk, which is higher than the 10.6% registered in the remaining parts of Greenland (*p* < 0.001). A higher proportion of patients in Nuuk were vaccinated against pneumococcus (36.3%) compared to patients in the remaining parts of Greenland (27.1%) (*p* = 0.003).

When looking at gender, significantly more men had debut time of COPD registered compared to women (35.3% vs. 25.7%, *p* = 0.002). No difference among men and women was observed when looking at other quality-of-care indicators.

## 4. Discussion

The prevalence of patients aged 20–79 years diagnosed with COPD in Greenland was found to be 2.2%. Among seniors aged 40 years or above, the prevalence was 3.8%. The prevalence was higher in Nuuk compared to patients from the remaining parts of Greenland and was higher among women compared to men. The quality of care was higher among patients living in Nuuk than among patients living in the remaining parts of Greenland. No difference in quality of care was observed among men and women.

### 4.1. Prevalence

In our study, we found a prevalence of patients aged 20–79 years diagnosed with COPD of 2.2% and the prevalence of diagnosed COPD among seniors aged 40 years or above to be 3.8%. In our study from 2019, only 152 patients were diagnosed with COPD, yielding a prevalence of 0.4% [7]. The observed increase in prevalence can be explained by increased diagnostic activity.

In our study from 2019, we estimated a total prevalence of 7.5% for patients aged 20–79 years and 9.7% for patients 40–79 years [7], thus showing a significantly higher prevalence than estimated in this study. However, this was expected, since the previous study estimated the prevalence of patients based on medication targeting obstructive lung disease (including both asthma and COPD), whereas we here estimated the prevalence of patients registered with a COPD diagnosis. Another study from 2016 included patients in Greenland aged 50+ using medication targeting obstructive lung disease (including both asthma and COPD) and found a prevalence of 7.9% [9]. However, only 34.8% had a spirometry performed within two years indicating medical treatment without exact diagnosis. Thus, patients with permanent obstruction may only represent a subgroup of the patients treated with medication targeting obstructive lung disease.

A Greenlandic study found lung function among Inuit to be higher than among Caucasians, with both FEV1 and FVC above 100% of the predicted values [18]. This has also been shown in other studies among Canadian and Greenlandic Inuit [19,20].

In 2021, an initiative was enrolled in Nuuk to increase use of diagnosis coding verified by spirometry. It was therefore not surprising to us that more diagnosed patients were found in Nuuk compared to the remaining parts of Greenland. As the initiative continues to be implemented in the remaining parts of Greenland in the following years, we expect the prevalence of patients diagnosed with COPD to increase. The current prevalence estimated in this study might thus be underestimated; however, it is the best estimate so far of the prevalence of COPD in Greenland.

We found significantly more women than men to be diagnosed with COPD, contradicting the findings by Safiri et al., who found the number of cases globally to be higher among men compared to women in patients aged 20–74 years [4]. The higher proportion of women diagnosed with COPD in our study might be due to women being more frequent users of the health care system, thereby increasing their chance of being diagnosed. In accordance, a study from Greenland showed that women aged 20–59 years more frequently were in contact with the Greenland health care than men [21]. This finding may also explain the underdiagnosing of men due to their significantly impaired lung function compared to women, as seen from a lower FEV1 of expected as well as FEV1/FVC ratio. The reduced lung function among men in our study might be explained by the fact that men probably go to the doctor when they experience more symptoms compared to women, so men will be diagnosed later. The fact that women are more frequently in contact with the health care system compared with men is a well-known phenomenon primarily due to masculine norms [22].

A study from Canada in 2018 including adult Inuit aged 18 years or above in Ottawa showed that COPD is one of the most common chronic diseases in the Canadian population, with a prevalence of 6.7% [23]. 

In our study, we observed that the prevalence of COPD increased with age. This is in agreement with a study from 2017 investigating the association between aboriginal people in Canada and COPD [24]. They further found the prevalence of COPD for patients aged 55 years or above to be 11.1%, which is considerably higher than our findings. However, their data were self-reported and not based on diagnosis criteria. They also found COPD to be strongly associated with aboriginal people, who were older, smokers, had a low socioeconomic status, and did not have direct access to health care when needed. However, when they compared Métis, First Nations, and Inuit, they observed the lowest prevalence of self-reported COPD among Inuit (3.21%) and the highest prevalence among Métis (7.93%). However, this might not be equally comparable to the Inuit population in Greenland, since the ethnicity has not been differentiated, indicating that a part of our study population may be on Scandinavian or Philippine inhabitants. For a future study, it would be interesting to study the association between socioeconomic status and COPD in Greenland, since the abovementioned study found an inverse association between income and COPD [24]).

To sum up, we expect the prevalence of COPD in Greenland to be low due to three main causes. (1) A number of chronic diseases are underdiagnosed in Greenland because of the general challenges in the Greenlandic health care center: large distances, lack of permanent staff, and short-term temporary workers focusing on reaction to acute conditions rather than proactive detection and treatment of chronic conditions [10,25]). (2) Until 2020, we had an inadequate setup with a spirometer, where we were not able to get mouthpieces for certain periods. (3) Finally, there is no Inuit reference for Greenlanders. Greenlanders have higher LFUs than expected compared to a European of the same height, age, and sex, thereby causing underdiagnosis [26]. 

### 4.2. Quality of Care

In the present study, quality of care was evaluated according to international guidelines [1]) and guidelines from SDCG [13].

Eight out of ten indicators of quality of care were higher among patients living in Nuuk compared to patients living in the remaining parts of Greenland, indicating higher quality of care in Nuuk. This, however, also reflects the increased diagnostic focus initiated by SDCG in Nuuk in 2021, which has positively affected many of the indicators of quality of care.

However, there is room for improvement in both Nuuk and the remaining parts of Greenland, as none of the quality-of-care indicators yet reaches the goals or suggestions from national and international guidelines.

The indicator with the highest registration rate was spirometry performed within the previous two years (84.3%), which is close to both national and international guidelines at >85.0% [13,27,28]. Registration of smoking status was 63.9% and did not reach the goal from national guidelines at >85.0%. In comparison, the quality of care among COPD patients from 82 different general practices in Denmark showed that 75.8% of the patients had their smoking status registered, while approximately 40% had a spirometry performed within one year [29].

We found that only 23.0% had a CAT score registered. However, the use of CAT questionnaires was only initiated in Nuuk in March 2022 and had started to be implemented in the remaining parts of Greenland in the summer of 2022, and the proportion of this indicator is thus expected to increase over the coming years. The low registration of patients vaccinated against pneumococcus (31.0%) might be explained by the fact that the pneumococcus vaccine has only been offered since 2020.

Smoking is a major risk factor for the development and progression of COPD, and smoking cessation is the only way to stop the progression of COPD, as also suggested by the guidelines [10,13,25]. In our study however, we found an extremely high prevalence of current smokers among patients diagnosed with COPD (65.3%). In fact, we found that more patients with COPD smoke on a daily basis compared to the general population in Greenland, where around half the population are current smokers [6,30]. This is in accordance with other studies showing that the prevalence of current smokers is higher among COPD populations compared to healthy groups [31,32,33]. Our findings further underline the need for an increased focus on smoking cessation in Greenland among COPD patients and in general. In a previous study, Backe and Pedersen investigated the effect of a diabetes initiative and found that the prevalence of smokers in the diabetes population decreased significantly following the initiative [34].

In conclusion, the diagnostic initiative has improved the quality of care associated with COPD due to a focused effort, similar to initiatives targeting other diseases, e.g., diabetes [10].

### 4.3. Strengths and Limitations

A major strength of this study is that it covers 95% of the population in Greenland aged 20 years or above. Furthermore, no previous study has investigated the prevalence and quality of care of COPD based on the medical diagnosis codes ICPC-2 and ICD10. 

The prevalence of COPD has increased since 2019, indicating increased use of spirometry and diagnosis coding, most likely a consequence of the initiative led by SDCG. 

Data were extracted from the EMR based on ICPC-2 and ICD10 codes. This is therefore the first study on COPD including the entire Greenlandic population based on diagnostic coding. 

Globally, an underestimation of approximately 50% of the total prevalence of patients with COPD is expected (3), indicating that the prevalence in Greenland is underestimated. Nevertheless, it must be assumed that patients most seriously affected by COPD have been diagnosed.

Our results show that more people in the capital Nuuk are diagnosed with COPD compared to people living in the remaining parts of Greenland, indicating that the diagnostic initiative is not fully implemented in the remaining parts of Greenland and that the prevalence of COPD is underestimated. 

Realistically, the prevalence is, however, presumed to be underestimated, especially among people living outside Nuuk. In addition, the prevalence of people aged 80 years or above is expected to be underestimated, since fewer have had the opportunity to participate, and few agreed to participate in the initiative.

## 5. Conclusions

In conclusion, the prevalence of diagnosed COPD in Greenland among people aged 20–79 years was 2.2%, which is lower than in other comparable populations. More women than men were diagnosed with COPD. The lower prevalence outside Nuuk indicates that COPD is underdiagnosed in Greenland. The associated quality of care did not meet the criteria of National and International guidelines but was higher among patients in Nuuk compared to patients from the remaining parts of Greenland. Continued focus on early detection of new cases and initiatives to improve and expand monitoring of quality-of-care measurements including both additional clinical and patient reported outcomes are recommended.

## Figures and Tables

**Table 1 ijerph-20-05624-t001:** Quality indicators.

Process Indicators
Percentage of patients in whom smoking status was assessed within the previous year *^,^ **
Percentage of patients in whom blood pressure was measured within the previous year
Percentage of patients in whom nutritional status (height, weight) was measured within the previous year **
Percentage of patients in whom a spirometry was performed within the previous two years *^,^ **
Percentage of patients who has been vaccinated against influenza within the previous two years *
Percentage of patients who has been vaccinated against pneumococcus *
Percentage of patients in whom degree of physical activity is determined within the previous two years *
Percentage of patients in whom a COPD grade (A,B,C,D) was registered within the previous two years *
Percentage of patients in whom a debut time of COPD was registered
Percentage of patients in whom a COPD assessment score has been registered

* Quality indicators proposed by [17]; ** Quality indicators proposed by [16].

**Table 2 ijerph-20-05624-t002:** Basic characteristics of patients diagnosed with COPD in Greenland in 2022.

Basic Characteristics of Patients Diagnosed with COPD in Greenland in 2022							
	Men		Women			Total	
	N	Mean (SD)	N	Mean (SD)	*p*	N	Mean (SD)
Age (years)	414	64 (8.5)	470	64 (10.0)	0.203	884	64 (9.3)
Weight (kg)	402	79 (20.9)	455	66 (18.9)	**<0.001**	857	72 (20.9)
Height (cm)	400	170 (8.4)	454	155 (7.5)	**<0.001**	854	162 (10.8)
BMI (kg/m^2^)	397	27 (6.5)	452	27 (7.1)	0.809	849	27 (6.8)
Blood pressure, systolic (mmHg)	355	134 (16.0)	397	133 (16.0)	0.263	752	134 (16.0)
Blood pressure, diastolic (mmHg)	356	82 (11.5)	397	78 (10.7)	**<0.001**	753	80 (11.3)
FEV1 (L)	377	2.2 (0.8)	419	1.6 (0.6)	**<0.001**	796	1.9 (0.7)
FEV1 (%)	375	68 (22.3)	418	74 (22.0)	**<0.001**	793	71 (22.3)
FVC (L)	375	3.6 (1.1)	413	2.5 (0.7)	**<0.001**	788	3.0 (1.1)
FVC (%)	375	87 (22.9)	416	89 (21.3)	0.257	791	88 (22.0)
FEV1/FVC-ratio (%)	378	59 (11.9)	417	64 (10.7)	**<0.001**	795	62 (11.5)
COPD assessment score (CAT)	101	17 (8.3)	102	17 (8.1)	0.421	203	17 (18.2)
	**N**	**Prevalence (*n*)**	**N**	**Prevalence (*n*)**	** *p* **	**N**	**Prevalence (*n*)**
Current smokers (%)	389	64.0 (249)	437	66.4 (290)	0.479	826	65.3 (539)

*N* = number of patients, *SD* = standard deviation, *p* = *p*-values, *n* = number of patients with positive answer; *p*-values below 0.05 are in bold.

**Table 3 ijerph-20-05624-t003:** The prevalence of patients diagnosed with COPD in Greenland in 2022.

	Nuuk (*n*/N) (95% CI)	The Rest of Greenland (*n*/N) (95% CI)	*p*	Men (*n*/N) (95% CI)	Women (*n*/N) (95% CI)	*p*	Total
Total prevalence (%) 20–79 years	2.4 (2.16–2.66) (348/14,428)	2.0 (1.82–2.17) (491/24,649)	**0.006**	1.9 (1.74–2.11) (397/20,630)	2.4 (2.18–2.62) (442/18,447)	**0.001**	2.2 (2.00–2.29) (839/39,077)
Prevalence among seniors (40+ years)	4.5 (4.08–4.99) (361/7958)	3.4 (3.12–3.69) (518/15,214)	**<0.001**	3.3 (3.02–3.65) (413/12,395)	4.3 (3.94–4.71) (466/10,777)	**<0.001**	3.8 (3.55–4.04) (879/23,172)
**Prevalence in age groups (%)**							
20–39 years	0.05 (0.00–0.10) (3/6597)	0.02 (0.00–0.05) (2/9929)	0.359	0.01 (0.00–0.03) (1/8504)	0.05 (0.00–0.10) (4/8022)	0.159	0.03 (0.00–0.06) (5/16,526)
40–59 years	2.4 (1.96–2.77) (127/5367)	1.6 (1.34–1.86) (142/8860)	**<0.001**	1.5 (1.22–1.75) (115/7743)	2.4 (2.00–2.75) (154/6484)	**<0.001**	1.9 (1.67–2.11) (269/14,227)
60–79 years	8.9 (7.73–9.97) (218/2464)	5.9 (5.32–6.53) (347/5860)	**<0.001**	6.4 (5.69–7.14) (281/4383)	7.2 (6.40–8.01) (284/3941)	0.162	6.8 (6.27–7.35) (567/8324)
80+ years	12.6 (6.83–18.37) (16/127)	5.9 (3.80–7.94) (29/494)	**0.009**	6.3 (3.41–9.23) (17/269)	8.0 (5.13–10.78) (28/352)	0.436	7.3 (5.21–9.29) (45/621)

95% CI = 95% confidence intervals, *n*/N = number of patients/population, *p* = *p*-values; *p*-values below 0.05 are in bold.

**Table 4 ijerph-20-05624-t004:** The quality of care among patients in Greenland diagnosed with COPD in 2022.

	Nuuk (95% CI) (n/N)	The Rest of Greenland (95% CI) (*n*/N)	*p*	Men (*n*/N) (95% CI)	Women (*n*/N) (95% CI)	*p*	Total (95% CI) (*n*/N)
**Process indicators within the previous year, % (*n*/N)**							
Patients in whom smoking status was assessed	72.5 (67.94–77.11) (264/364)	57.9 (53.64–62.18) (301/520)	**<0.001**	61.6 (56.91–66.28) (255/414)	66.0 (61.67–70.24) (310/470)	0.178	63.9 (60.75–67.08) (565/884)
Patients in whom blood pressure was measured	59.1 (54.01–64.12) (215/364)	47.9 (43.59–52.18) (249/520)	**0.001**	52.2 (47.36–56.99) (216/414)	52.8 (48.25–57.28) (248/470)	0.860	52.5 (49.20–55.78) (464/884)
Patients in whom nutritional status (height, weight) was measured	74.7 (70.26–79.19) (272/364)	68.1 (64.07–72.08) (354/520)	**0.032**	67.9 (63.38–72.37) (281/414)	73.4 (69.41–77.40) (345/470)	0.071	70.8 (67.82–73.81) (626/884)
**Process indicators within the previous two years, % (*n*/N)**							
Patients in whom a spirometry was performed	92.9 (90.21–95.50) (338/364)	78.3 (74.72–81.81) (407/520)	**<0.001**	86.2 (82.91–89.55) (357/414)	82.6 (79.12–85.98) (388/470)	0.134	84.3 (81.88–86.68) (745/784)
Patients vaccinated against influenza	54.4 (49.28–59.51) (198/364)	56.0 (51.69–60.23) (291/520)	0.645	55.1 (50.28–59.86) (228/414)	55.5 (51.04–60.02) (261/470)	0.891	55.3 (52.04–58.59) (489/884)
Patients in whom degree of physical activity was determined	45.3 (40.22–50.44) (165/364)	51.0 (46.66–55.26) (265/520)	0.099	48.1 (43.25–52.88) (199/414)	49.2 (44.63–53.67) (231/470)	0.748	48.6 (45.35–51.94) (430/884)
Patients registered with a COPD grade (A,B,C,D)	60.2 (55.14–65.19) (219/364)	16.5 (13.35–19.73) (86/520)	**<0.001**	37.9 (33.25–42.60) (157/414)	31.5 (27.29–35.69) (148/470)	0.044	34.5 (31.37–37.64) (305/884)
**Process indicators, % (n/N)**							
Patients registered with a debut time of COPD	53.6 (48.45–58.69) (195/364)	13.9 (10.88–16.81) (72/520)	**<0.001**	35.3 (30.66–39.87) (146/414)	25.7 (21.79–29.70) (121/470)	**0.002**	30.2 (27.18–33.23) (267/884)
Patients registered with a COPD assessment score	40.7 (35.61–45.71) (148/364)	10.6 (7.93–13.22) (55/520)	**<0.001**	24.4 (20.26–28.53) (101/414)	21.7 (17.98–25.43) (102/470)	0.342	23.0 (20.19–25.74) (203/884)
Patients vaccinated against pneumococcus	36.3 (31.41–41.27) (133/364)	27.1 (23.29–30.94) (141/520)	**0.003**	31.6 (27.16–36.12) (131/414)	30.4 (26.27–34.59) (143/470)	0.696	31.0 (27.95–34.04) (274/884)

95% CI = 95% confidence intervals, *n*/N = number of patients/population, *p* = *p*-values. *p*-values below 0.05 are in bold.

## Data Availability

Due to the nature of the research and due to ethical and legal supporting data is not available.

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
