# Peer review of "Low Prevalence of Chronic Obstructive Pulmonary Disease in Greenland—A Call for Increased Focus on the Importance of Diagnosis Coding"

_ijerph, 2023, doi:10.3390/ijerph20095624_

Round 1

Reviewer 1 Report

In this cross-sectional study, the authors evaluated the prevalence of COPD according to age and residence among COPD patients in Greenland as well as quality of care for these patients. The authors used data from a cross-sectional study. The topic is interesting and adds further knowledge on epidemiological distribution of the disease in Greenland. The prevalence of COPD was found to increase with age and be different across men and women; patterns of quality of care differed according to the geographic location (the capital vs the other parts of Greenland), thus underlying the need of standardizing diagnosis in order to tailor targeted therapeutic interventions in at risk groups.

The study is well written, I have only some minor changes to suggest:

-at line 36 change "vary" with "varies".

-at line 48 adjust redundant phrase "lack of diagnostic focus on diagnosing".

-at line 55 add "the" before "Steno..".

-at line 295 change "has improve" with "has improved".

Reviewer 2 Report

Well-written paper showing the healthcare needs of Greenland island. Few points that could be worked on: 

1. Since the flu vaccine is an annual vaccine, does vaccination in the last 2 years count as a good process indicator?

2. Lines 159-161 - do you mean to say compared to patients living in remaining parts of Greenland?

3. Lines 207 - observed increase in prevalence - any impact of the pandemic on this increased prevelance? Since you compare the one from 2019 versus post covid pandemic, during which healthcare utilization would have been more?

4. Please highlight the plausible reasons why women are more frequent users of the healthcare system than men, based on evidence. 

5. Any data on how many patients were on smoking cessation medications?

Reviewer 3 Report

Low FEV1.FVC has different significances between under 40 y.o. and over 40 y.o. It may not be appropriate to diagnose people under 40 y.o. as COPD. 

Why is the prevalence of COPD low inspite  of high prevalence of smoking?

What are the reasons for undergoing pulmonary function tests in the people? Are they symptomatic?

Why were more women diagnosed as COPD compared with men?

Why did men show more reduced lung function compared with women?

The examination of circumstances in young people with airflow obstruction would broaden our insight into the impact of smoking in younger ages.

 The combined observation of use of bronchodilator use would be interesting.

What kind of intervention wold improve the medical management of COPD in Greenland?

Is the literacy of people on COPD sufficient in Green land?

Are there different motivation for performing pulmonary function tests in doctors? 

Round 2

Reviewer 3 Report

The main reasons or criteria to perform pulmonary function tests should be added to the Methods section to offer the appropriate interpretation of the current study.  

Author Response

Dear reviewer, 

Thank you for this great comment. We have added criteria for performing pulmonary function test to the method section. Please see 103-106.